# The mouse metallomic landscape of aging and metabolism

Jean-David Morel [1,9], Lucie Sauzéat [2,5,6,9], Ludger J. E. Goeminne [1], Pooja Jha[1], Evan Williams [1,7], Riekelt H. Houtkooper[1,8], Ruedi Aebersold [3,4], Johan Auwerx [1✉] & Vincent Balter [2✉]

Organic elements make up 99% of an organism but without the remaining inorganic bioessential elements, termed the metallome, no life could be possible. The metallome is involved in all aspects of life, including charge balance and electrolytic activity, structure and conformation, signaling, acid-base buffering, electron and chemical group transfer, redox catalysis energy storage and biomineralization. Here, we report the evolution with age of the metallome and copper and zinc isotope compositions in five mouse organs. The aging metallome shows a conserved and reproducible fingerprint. By analyzing the metallome in tandem with the phenome, metabolome and proteome, we show networks of interactions that are organ-specific, age-dependent, isotopically-typified and that are associated with a wealth of clinical and molecular traits. We report that the copper isotope composition in liver is age-dependent, extending the existence of aging isotopic clocks beyond bulk organic elements. Furthermore, iron concentration and copper isotope composition relate to predictors of metabolic health, such as body fat percentage and maximum running capacity at the physiological level, and adipogenesis and OXPHOS at the biochemical level. Our results shed light on the metallome as an overlooked omic layer and open perspectives for potentially modulating cellular processes using careful and selective metallome manipulation.

[1] Laboratory of Integrative Systems Physiology, Institute of Bioengineering, Ecole Polytechnique Fédérale de Lausanne, Lausanne 1015, Switzerland. [2] Université de Lyon, Ecole Normale Supérieure de Lyon, Université de Lyon 1, CNRS, LGL-TPE, Lyon, France. [3] Department of Biology, Institute of Molecular Systems Biology, ETH Zürich, Zürich, Switzerland. [4] Faculty of Science, University of Zürich, Zürich, Switzerland. [5] Present address: Université Clermont Auvergne, CNRS, Inserm, Génétique, Reproduction et Développement, F-63000 Clermont-Ferrand, France. [6] Present address: Université Clermont Auvergne, CNRS, IRD, OPGC, Laboratoire Magmas et Volcans, F-63000 Clermont-Ferrand, France. [7] Present address: Luxembourg Centre for Systems Biomedicine, University of Luxembourg, Esch-sur-Alzette, Luxembourg. [8] Present address: Laboratory Genetic Metabolic Diseases, Amsterdam UMC, University of Amsterdam, Amsterdam, The Netherlands. [9] These authors contributed equally: Jean-David Morel, Lucie Sauzéat. ✉email: admin.auwerx@epfl.ch; vincent.balter@ens-lyon.fr

Knowledge of the metallome – briefly, the set of inorganic elements in an organism, (for a complete definition see Lobinski et al.[1]) and its interactions with other omic layers is elemental for understanding how physiological perturbations cause or result from alterations of metal concentrations and stable isotope compositions. Isotopic fractionation, the partitioning of stable isotopes due to slight differences in mass-dependent quantum zero-point energies[2,3], is an emerging means for characterizing the involvement of the metallome in biological processes[4–6]. So far, interactions of the metallome with other omic layers have been mostly studied in plants to optimize crop production[7–9]. Scarcer studies in yeast[10] and cultured human cells[11] reveal a wealth of metallome-dependent biological traits. The present study is the first comprehensive analysis of the interactions between the metallome and other omic layers (phenome, metabolome, and proteome) in organs of mice as a function of age.

## Results and discussion

**The metallome and isotope compositions fingerprints are organ-specific.** We measured the metallome (K, Mg, Na, P, S, Ca, Fe, Cu, Rb, Zn, Se, Co, Mo, and Cd) and the Cu and Zn stable isotope compositions (denoted hereafter $\delta^{65}$Cu and $\delta^{66}$Zn, respectively) in five organs (brain, heart, kidney, liver, and muscle, Supplementary data 1 and table S1) of 49 C57BL/6 male mice aged 6-, 16-, and 24-months old (mo) (Fig. 1a). Principal Component Analysis (PCA) shows organ-specific metallomic and isotopic signatures (Fig. 1b) as described before[12–14]. The first principal component is driven oppositely by protein-bound transition metals (Fe, Zn, Se, Mn, Co, and Cu) and free alkali and alkaline earth metals (Ca, K, and Mg) (Fig. 1b). The $\delta^{65}$Cu and $\delta^{66}$Zn values are also organ-specific, suggesting that isotope fractionation depends on fine, organ-specific processes (Fig. 1c, d). The $\delta^{66}$Zn value in heart, brain, muscle and liver follows a linear relationship with Zn concentration, a relationship that is known to indicate identical Zn routing in these organs[15], obviously distinct in kidneys where glomerular filtration probably drives isotopic fractionation through a distillation process[16] (Fig. 1c). Copper isotope fractionation is more intense than that of Zn. Cu exhibits a unique concentration vs isotopic composition signature for each organ, depending on the occurrence of its oxidation state, Cu(II) compounds being isotopically heavier than Cu(I) compounds[17] (Fig. 1d).

We further explored the metallome evolution during aging. Major elements, i.e. electrolytes (K, Mg, and Na) or molecular constituents (S and P), show a remarkable homogeneity among organs and stability over time (Fig. S1A). Minor elements (Ca, Cu, Fe, Rb, Zn), with countless biological roles[18], and especially ultra-trace elements – Se for selenoproteins, Co for vitamin B12, Mo for molybdoenzymes, and Cd with unknown biological role[18] – exhibit a more heterogeneous distribution in the body than major elements and vary with age (Fig. S1B–S1C). Such ultra-trace elements are mainly present in the liver and to a lesser extent in the kidney and brain. The gastrocnemius striated muscle and the myocardial muscle are depleted in ultra-trace elements, probably reflecting their limited involvement in specific biochemical synthesis (Fig. S1C).

**The metallome and isotope compositions fingerprints are age-dependent.** Of all the organs, metal concentrations in the brain seem to vary the most with age (Fig. 1e, Rb, Fe, Ca, Cu, and Co at $p$-value < 0.05). We find an age-dependent accumulation of Fe and Cu in a healthy brain (Fig. 1f), which complements previous observations[19–22] in the mouse. As high levels of transition metals such as Fe, Cu, and Zn are known to be associated with amyloid-β

plaques and α-synuclein accumulation in neurodegenerative diseases[23,24], all these results may suggest that metal accumulation precedes the formation of the protein aggregates. The accumulation of Cd in kidney (Fig. 1f) may be illustrative of chronic toxicity[25]. Rubidium, which is the K minor element analog[26], decreases between 6 m.o. and 24 m.o. in brain, muscle, and kidney (Figs. 1f and S2), suggesting unprecedented evidence of disrupted K metabolism in old mice. These results confirm the value of the metallome as a biomarker for aging[14], but our data extend this observation to isotope compositions, notably the decrease of the liver $\delta^{65}$Cu value during aging (Fig. 1f). Isotopic aging "clocks" have been described in yeast (D/H, $^{13}$C/$^{12}$C, $^{15}$N/$^{14}$N,[27]), worm ($^{65}$Cu/$^{63}$Cu,[28]), and in human blood ($^{65}$Cu/$^{63}$Cu, $^{66}$Zn/$^{64}$Zn[29]). Furthermore, metallomic or isotopic aging clocks are much easier to handle than DNA methylation clocks (performed for a subset of livers of the same animals[30]) which require a predetermined profile of methylation sites in the organism and organ of interest. To find metal correlations not solely driven by age, we hereafter used linear regression to correct metallomic concentrations for age effects. This correction is applied hereafter except if specified otherwise and ultra-trace elements are excluded to compare organs with the same set of metals. The concentration of major elements Mg, K, P, and S, but not Na, are positively and strongly correlated in all organs and conserved after age correction, highlighting the fundamental interplay throughout life between cationic electrolytes ($Mg^{2+}$ and $K^+$), ATP ($PO_4$) and glutathione (S)[31] (Figs. 1g and S3). Minor elements Ca and, to a lesser extent Fe, are negatively correlated to other elements in many organs (Fig. 1g).

**The metallome fingerprint is highly conserved across different studies.** We compared our results with a recent mouse study[14], which measured a smaller subset of the metallome in mouse organs at different ages. With the same set of metals and organs, the respective PCAs agree almost perfectly, demonstrating a highly conserved metallome distribution in mouse organs (Fig. 2a, b). For a given organ, age effects converged in both studies despite very different timepoints (16 timepoints at 2 months intervals[14] versus 3 timepoints in this study) and different diets (NIH-31[14] vs Teklad 18% protein) (Fig. 2c). After age correction, the correlation networks generated with both datasets still converge in most organs (Figs. S4 and 2d). This suggests a tight control of the metallome, reflective of the duality of metals, being bioessential at normal concentrations and toxic outside. From an evolutionary perspective, the metallome homeostasis likely evolved billions of years ago when early life had to adapt to an inorganic environment driven by ocean chemistry[32]. Our work reveals that not only basal metal concentrations in organs are conserved across unrelated studies, but those subtle variations are also conserved, as evidenced by the similar interaction network and reproducible response to aging. Given this, we asked whether these metal variations would relate to perturbations in other biological layers, notably the phenome, metabolome, and proteome.

**The metallome and isotope compositions are associated with metabolic health and mitochondrial content.** We recorded cardiometabolic phenotypic traits, including body weight, fat content, $O_2$ consumption, glucose level, and blood pressure, and compared these with the metallome and $\delta^{65}$Cu and $\delta^{66}$Zn profiles through pairwise correlations (Figs. 3a, b and S5). Amongst the metallome-phenotype associations, the most significant is a negative correlation between the liver $\delta^{65}$Cu value and body weight (Fig. 3b), and to a lesser extent fat% and a slower return to glucose homeostasis after intraperitoneal injection (IPGTT AUC,

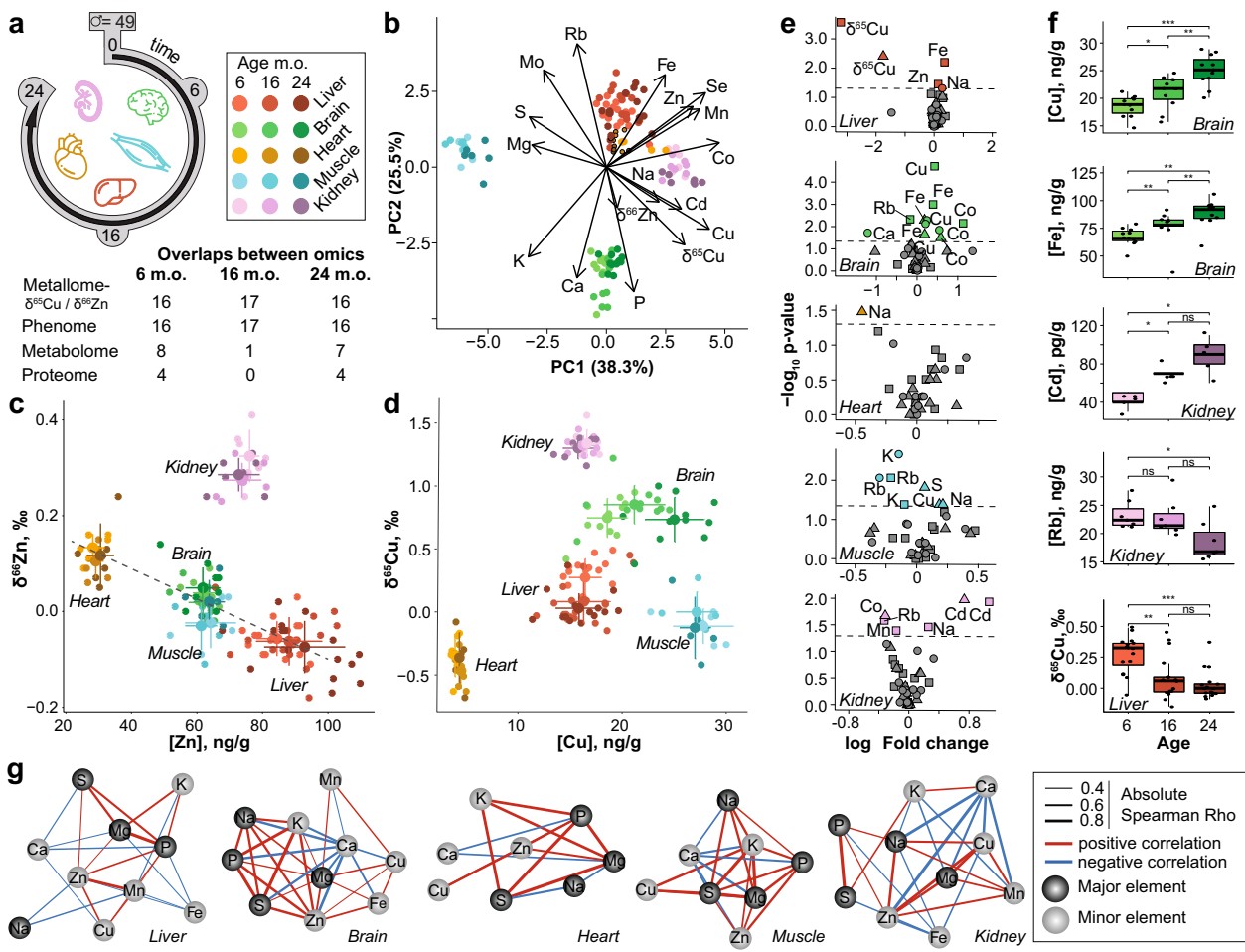

**Fig. 1 Organs have distinct metallomic fingerprints, that evolve with aging. a** Study design and overlapping samples between omic layers. Icons are designed by www.freepik.com, and assembled by VB on Adobe Illustrator. **b** Principal component analysis of the metal concentration and isotope composition of each sample. Clear organ signatures are visible, with small shifts with aging. **c**, **d** Relation between isotope composition and concentrations of Cu (**c**) and Zn (**d**). Error bars indicate standard deviation. The dashed line indicates a linear relationship between $\delta^{66}$Zn and Zn between the liver, brain, muscle, and heart, indicative of identical Zn routing in these organs. **e** Volcano plots of the effect of aging on metal concentrations. Triangles: 16 vs 6 m.o., circles 16 vs 24 m.o., squares: 24 vs 6 m.o. The dotted line indicates the threshold of Adj.$P < 0.05$, FDR-corrected (limma) moderated $t$-test, two-sided. **f** Boxplot of the most age-dependent metal concentrations and isotope compositions. The lower and upper hinges correspond to the first and third quartiles, and center line is the median. The whiskers extend from the hinge to the largest value no further than 1.5 * inter-quartile range. ***Adj.$P < 0.001$, **Adj.$P < 0.01$, * Adj.$P < 0.05$, FDR-corrected (limma) moderated $t$-test, two-sided. **g** Network of Spearman Rho correlations between metals. Only correlated metals with **Adj.$P < 0.001$, **Adj.$P < 0.01$, * Adj.$P < 0.05$, (limma) moderated $t$-test, with age as a covariate, are pictured. The correlation coefficient and directionality of the correlation are pictured through line thickness and color, respectively. Source data are provided as a Source Data file.

Fig. 3b), an indicator of diabetic-like symptoms. Iron and $\delta^{65}$Cu are further associated with an increase in the activity of mitochondrial complex I and a reduction of the mtDNA/nuclear DNA ratio in the liver, suggesting that liver mitochondrial activity is associated to these changes in metal levels. A high liver $\delta^{65}$Cu is therefore associated with both younger (Fig. 1f) and metabolically healthier (Fig. 3b) animals. While increased Cu concentration[33] and isotope composition[4,34,35] have been previously associated with metabolic activity and growth of tumors, our study shows that $\delta^{65}$Cu may be a more general indicator of metabolic fitness. Importantly, these liver metal concentrations may not be causal in metabolic fitness, but rather represent a biomarker of liver health in metabolically healthy animals, as opposed to mildly obese or diabetic animals which develop liver dysfunction. Increased liver Fe further correlated with fat percentage and muscle ROS production (Fig. 3c), which are produced primarily by mitochondria during muscle exercise[36]. In addition, liver calcium was strongly correlated with the activity of mitochondrial

complex IV. Calcium has been shown to bind and inhibit mitochondrial complex IV in vitro[37], but these results suggest that interactions between Ca and complex IV may also be relevant in vivo. Kidney Fe also correlated with respiratory capacity (VO$_2$ increase, Fig. 3c) reflecting the importance of Fe signaling in kidney erythropoietin production, a critical regulator of hemoglobin and erythrocyte production[38].

**The metallome and isotope compositions fingerprints integrate with overlapping biological pathways in the proteome and metabolome.** To examine the interplay between metallomics and other omics layers, we used previously published metabolomics[39] and new proteomics analyses performed on liver samples from the same animals. A Metabolite Set Enrichment Analysis (MSEA) and Gene Set Enrichment Analysis (GSEA) based on correlations between the metallome, the metabolome (Fig. 4a), and the proteome (Fig. 4b) define four groups of correlated metals. Group I

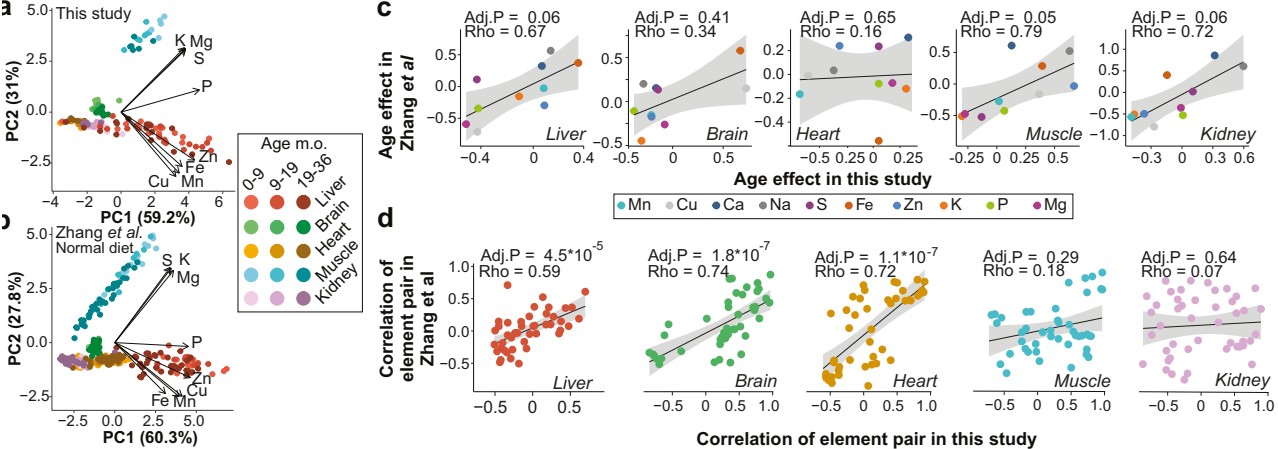

**Fig. 2 The metallome organ signature and its response to stimuli are conserved across datasets. a**, **b** Robustness of PCA analysis of the mouse metallome across different data sets. **a** PCA for the present study. **b** PCA analysis for the study of Zhang et al.[14] **c** Scatterplots of Spearman Rho correlation coefficients, and Spearman test FDR-corrected *p*-value calculated between metals and age in the data from Zhang et al. vs. those of the present study. The Zhang dataset measured metals at 16 timepoints at 2 months interval, as opposed to our studies' 3 timepoints, yet trends converge towards the same age effects in the liver, muscle, and kidney. The effects of age in the heart do not match, but are very low in both studies. **d** Scatterplots of Spearman Rho correlation coefficients and Spearman test FDR-corrected *p*-value calculated between pairs of metals from Zhang et al. vs. those of the present study. To enable comparisons between the two studies, data were normalized to Na in **a**, **b** and total metal concentrations in **c**, **d**. In panels **c** and **d**, the black line and gray error band represent a linear regression and its 95% confidence interval, respectively. Source data are provided as a Source Data file.

(S, Mg, P, Rb, K, Cu, Zn, and Mn) displays strong positive normalized enrichment scores (NES) for amino, organic, and carboxylic acids and derivatives and negative NES for fatty acids, lipids, and derivatives, with opposite enrichments in Group II (Na and Ca) (Fig. 4a). This pattern is independently confirmed by the GSEA on hallmark gene sets in the proteome, where groups I and II oppositely correlate with adipogenesis and fatty acid metabolism (Fig. 4b). The MSEA for cellular locations indicates that group I associates with cytoplasmic components but negatively correlates with membrane components (Fig. 4a). Group I also negatively correlates with the mitochondria (Fig. S8), in accordance with the Hallmark GSEA concerning the oxidative phosphorylation pathway (OXPHOS, Fig. 4b). Group III consists of $\delta^{65}Cu$ only and is positively associated for amino, organic and carboxylic acids and derivatives in the chemical classes MSEA (Fig. 4a). The previously observed negative phenotypic correlation between the $\delta^{65}Cu$ value and fat % (Fig. 3b) is strengthened by the Hallmark GSEA (Fig. 4b) implicating adipogenesis. Group IV (Fe and $\delta^{66}Zn$) had comparably fewer enrichments with the MSEA (Fig. 4a) but reveals many significant enrichments with the GSEA (Figs. 4b and S8). This group associates with mitochondrial content, in accordance with the role of Fe in heme synthesis, electron transport and oxidative phosphorylation, and the mitochondrial biosynthesis of the critical Fe-S cluster proteins[40,41]. The discovery that the $\delta^{66}Zn$ value varies in opposition to Zn concentration, notably with mitochondrial proteins (GO CC, Figs. 4b and S8), emphasizes the added value of determining stable isotope composition in addition to concentration to highlight metal-based molecular pathways. The KEGG GSEA (Fig. S8) links $\delta^{66}Zn$ and neurodegenerative diseases, predicting a connection that was experimentally confirmed in the APPswe/ PSEN1dE9 mouse model of Alzheimer disease[42,43].

We summarize the results of the study in a network analysis gathering correlations between metals, phenotypes, as well as the MSEA and GSEA results in the liver (Fig. 4c). The resulting network recapitulates all the above observations (e.g. $\delta^{65}Cu$ and body weight, Fe and OXPHOS) and shows that the metallome significantly interacts with all the studied omic layers. The network highlights the similar roles of K and Rb which strengthen the postulated status of Rb as a proxy for K metabolism.

**Perspectives**. Overall, our findings establish that the metallome and the isotopic compositions are biologically relevant and highly integrated omic layers with an unprecedented level of conservation in both basal signature and evolution with age. Subtle variations in metal concentrations are conserved across studies and associated with markers of metabolic health at both the physiological (body weight, fat percentage, and insulin resistance in the IPGTT) and biochemical levels (OXPHOS, lipid synthesis among others). A limitation of this study is that it is mostly based on correlations, and we do not imply that alterations in metals are causal in the phenotypes observed. However, the high reproducibility of metallomics suggests that changes in metal concentration may represent a source of reliable and affordable biomarkers of organismal, tissular, cellular, and subcellular aging and metabolism.

## Methods

**Mouse sample collection**. All animal experiments were performed according to Swiss ethical guidelines and approved by the local animal experimentation committee of the Canton of Vaud under license 2172. Male C57BL/6J mice (Janvier, St Berthevin, France) were fed with a standard chow diet (containing 18% protein, 50% carbohydrate, and 6.0% fat; Harlan Laboratories, Madison, WI, USA) at the Laboratory of Integrative Systems Physiology (LIPS) at EPFL (Ecole Polytechnique Fédérale de Lausanne). The mice were kept with a 12-h light cycle and temperature was regulated at 23 °C and 40–60% humidity. After in vivo phenotyping, organs (i.e. liver, brain, heart, muscle (i.e. quadriceps), and kidney) were collected at three different time-points in their life i.e. 6 months (mature adult), 16 months (middle age) and 24 months (old). The organs were flash frozen in liquid nitrogen ($N_2$), and stored at −80 °C, to promote their conservation before chemical and metabolomics analyses.

**In vivo phenotyping**. All phenotype data used in this paper was previously published in Houtkooper et al.[39], but a brief description of the procedures is provided here. Body composition was determined by Echo-MRI (Echo Medical Systems, Houston, TX, USA) and oxygen consumption ($VO_2$), respiratory exchange ratios (RER), activity levels and activity were monitored by indirect calorimetry using the comprehensive laboratory animal monitoring system (CLAMS) (Columbus Instruments, Columbus, OH, USA). Glucose tolerance was analyzed by measuring blood glucose and insulin following intraperitoneal injection of 2 g/kg glucose after an overnight fast. Maximum respiration potential ($VO_2max$) was determined in a metabolic treadmill (Columbus instruments, Columbus, OH, USA) with an incremental speed protocol. During the run, $VO_2$ and $VCO_2$ are measured. The experiment is stopped when mice are exhausted, RER is above 1 for more than one

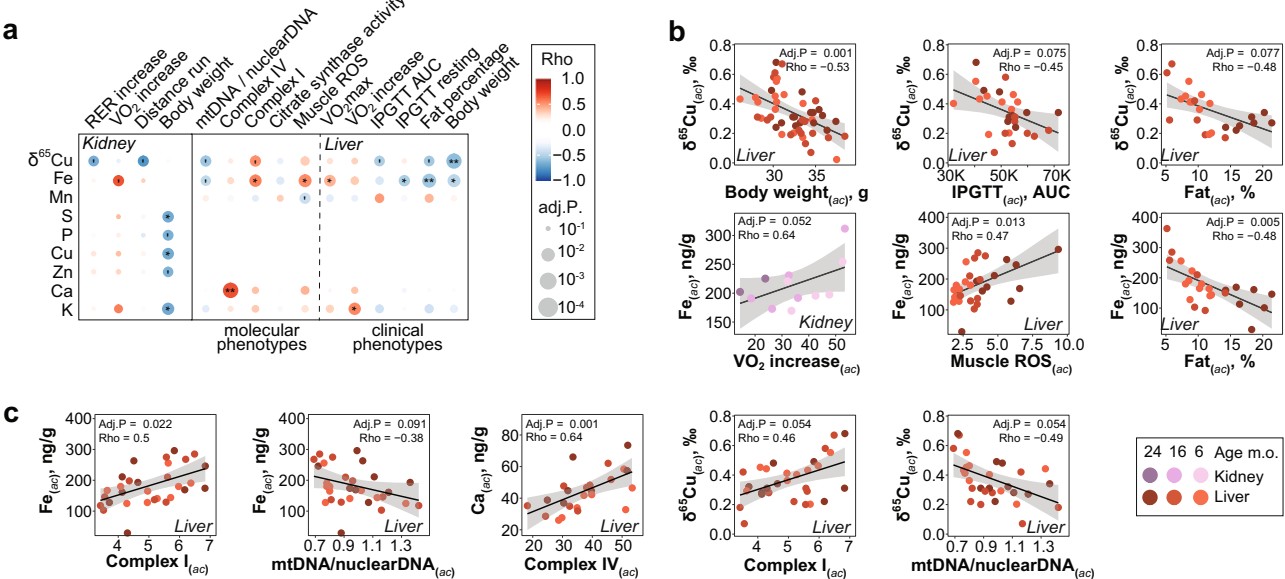

**Fig. 3 Metals and isotope compositions, notably Fe and $\delta^{65}$Cu in liver and kidney associate with metabolic traits. a** Significant correlations between metabolic phenotypes and metals in the liver and kidney. There were no significant associations in other organs except for a single one in the heart. This is consistent with the central role of liver and kidney in the control of metabolic traits. ***Adj.$P$ < 0.001, **Adj.$P$ < 0.01, *Adj.$P$ < 0.05, 'Adj.$P$ < 0.1. **b, c** Dot plots of the individual correlations of $\delta^{65}$Cu (**b**) and Fe (**c**) with indicators of obesity (body weight, fat %), diabetes (Intra-peritoneal glucose tolerance test, area under the curve, IPGTT AUC), and muscle activity (Muscle ROS) and running performance (VO$_2$ increase), as well as the activity of mitochondrial complex I. The subscripts (ac) stand for "age corrected". In all panels, Rho indicates the Spearman Rho coefficient, and Adj.$P$ is computed by a moderated $t$-test (limma), two-sided and FDR-corrected with age as a covariate. In panels **b** and **c**, the black line and gray error band represent a simple linear regression and its 95% confidence interval, respectively. Source data are provided as a Source Data file.

interval, or the VO$_2$ reaches a clear plateau. Liver mitochondrial complex I and IV activity (Mitosciences, Eugene, OR, USA), citrate synthase (Sigma) were determined according to the manufacturer's protocols. Liver and muscle ROS were determined by 4-HNE detection in PBS tissue lysates using an ELISA-based method (Cell Biolabs, San Diego, CA, USA).

**Analyses of major and trace element concentrations.** All the chemical analyses were carried out in a clean laboratory below laminar flow clean hoods using ultra-pure water at 18.2 MΩ.cm and acids that were doubly distilled to ensure low trace element contents and avoid any exogenous contaminations. Blanks were also run to quantify and integrate potential contamination from the material and/or acids used to perform experiments. The samples were crushed under liquid N$_2$, freeze-dried, weighted, and then dissolved in a concentrated HNO$_3$-H$_2$O$_2$ 30% mixture in Savillex beakers for about 72 h at 120 °C. Once dissolved, major (Ca, K, Na, P, S, and Mg) and trace elements (Mn, Co, Cu, Zn, Fe, Se, Rb, Mo, and Cd) were measured using the ICP-AES, iCAP 6000 Radial and the quadrupole ICP-MS Thermo iCap-Q respectively at the Ecole Normale Supérieure (ENS) of Lyon. The concentrations were calculated using calibration curves determined based on multi-elemental solutions (SP-33MS for trace element and in-house solution made from mono-elemental solutions for major element) following the procedure detailed in Garçon et al.[44]. Both accuracy and reproducibility were monitored by complete duplicate analyses, replication of an in-house (sheep plasma; OEP) and certified standards (bovine liver, SRM-1577c) as well as re-run analyses over the course of an analytical session. Given our long-term reproducibility on the in-house/certified standards and duplicate analyses, the 2 standard deviations (2 SD) analytical uncertainty adopted in this study for both trace and major concentrations are on average better than 10% (Tables S1 and S2).

**Analyses of copper and zinc isotopic compositions.** Copper and zinc isotopic compositions were measured at the Ecole Normale Supérieure (ENS) of Lyon following the procedure described by Maréchal et al.[45]. Briefly, before each isotopic measurement, the sample solutions were purified by ion-exchange chromatography using quartz columns filled with 1.8 mL of Bio-Rad AGMP-1 (100–200 mesh) anion-exchange resin. Both copper and zinc were successively eluted with 20 mL of HCl 7 N + 0.001% H$_2$O$_2$ and 10 mL of HNO$_3$ 0.5 M, respectively. On the day of the analyses, Zn and Cu purified solutions, previously evaporated to dryness, were dissolved in a Cu (Cu SRM 976, National Institute of Standards and Technology, Gaithersburg, MD, USA) or Zn-doped solution (Zn JMC 3-0749 L, Johnson Matthey Royston, UK) respectively to reach a Zn or Cu sample concentration of

about 300 ppb, which is similar to the concentration of the standard solution that was run between each sample.

Copper and Zn isotopic compositions are expressed as:

$$\delta^{65}Cu_{sample}(\text{‰}) = \left[\frac{\left(\frac{65Cu}{63Cu}\right)_{sample}}{\left(\frac{65Cu}{63Cu}\right)_{standard}} - 1\right] * 1000 \qquad (1)$$

and

$$\delta^{66}Zn_{sample}(\text{‰}) = \left[\frac{\left(\frac{66Zn}{64Zn}\right)_{sample}}{\left(\frac{66Zn}{64Zn}\right)_{standard}} - 1\right] * 1000 \qquad (2)$$

Copper and Zn isotopic compositions were measured on a Nu Plasma (Nu 500) MC-ICP-MS in wet plasma conditions. Zn JMC 3-0749L (also called JMC-Lyon; Johnson Matthey, Royston, UK) and Cu SRM 976 were used as reference standards for $\delta^{66}$Zn and $\delta^{65}$Cu, respectively. Instrumental mass fractionation was corrected with an exponential law using an elemental-doping method and instrumental drift over time was controlled with standard sample bracketing[45]

The accuracy of isotopic compositions was assessed by the analysis of in-house and commercial standard solutions (i.e. sheep plasma; OEP and bovine liver SRM-1577c certified reference material[46]) during each analytical sequence. For Zn, the average $\delta^{66}$Zn value measured is +0.76 ± 0.08‰ ($n = 3$, 2 SD) for OEP and −0.18 ± 0.05‰ ($n = 4$, 2 SD) for SRM-1577c which is in good agreement with the average values 0.73 ± 0.09‰ ($n = 17$, 2 SD) for OEP and −0.19 ± 0.06‰ ($n = 20$, 2 SD) for SRM-1577c (4). For Cu, we measured the OEP $\delta^{65}$Cu value at −1.10 ± 0.15‰ ($n = 3$, 2 SD) and the SRM-1577c $\delta^{65}$Cu value at +0.43 ± 0.04‰ ($n = 4$, 2 SD) which is also in good agreement with previously estimated values ($\delta^{65}$Cu$_{OEP}$ = −1.14 ± 0.13 ‰ (2 s, $n = 35$) and $\delta^{65}$Cu$_{1577c}$ = +0.37 ± 0.14 ‰ ($n = 19$, 2 SD, (4), Table S1). Based on re-run samples, complete duplicate analyses and standard measurements, we estimate the precision of our measurements at ±0.12‰ (2 SD). The long-term precision based on the repeated measurements of standard Zn JMC 3-0749L and Cu SRM 976 alone is however better than ±0.06‰ ($n = 140$, 2 SD).

**Metabolomics.** The liver metabolomics was performed by Metabolon (Durham, NC, USA) and has already been published[47]. Briefly, sample preparation was conducted using a proprietary series of organic and aqueous extractions to remove the protein fraction while allowing maximum recovery of small molecules. The extracted samples were split into equal parts for analysis on the GC/MS and LC-MS/MS platforms. For LC-MS/MS, samples were split in two aliquots that were

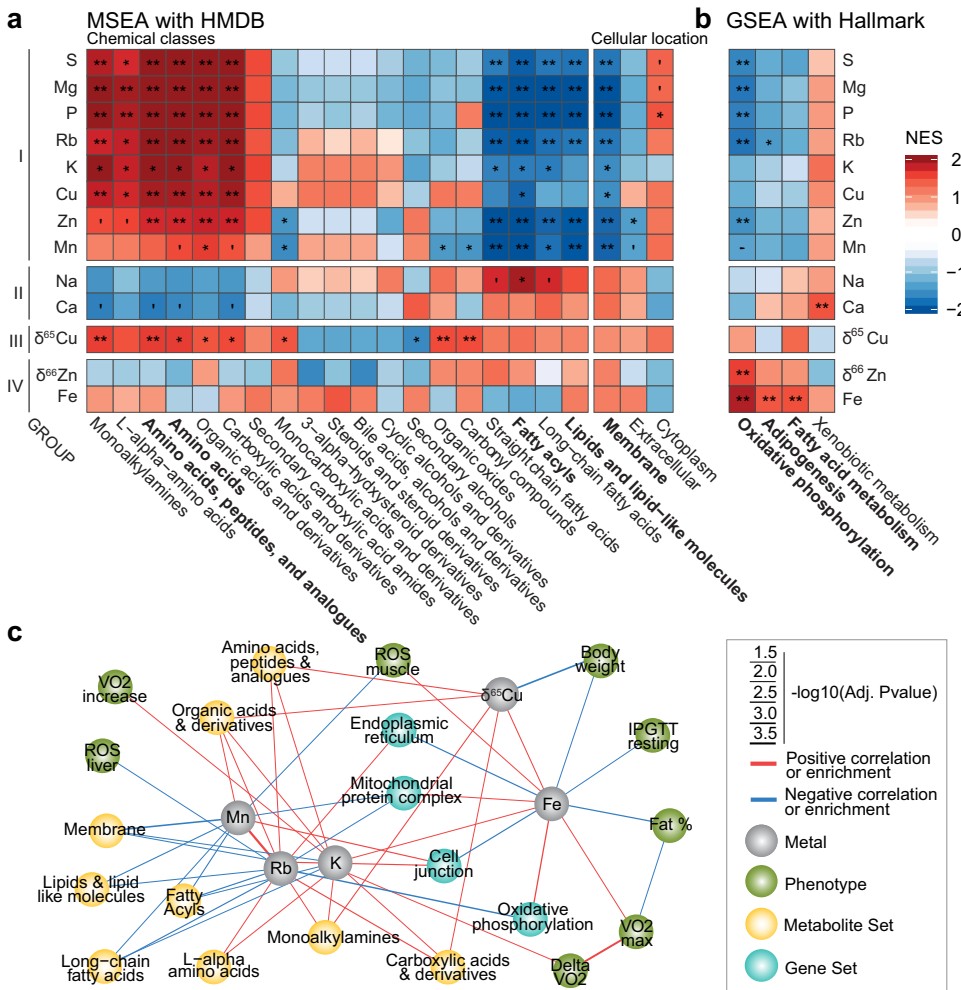

**Fig. 4 Enrichment analyses in the liver metabolome and proteome converge towards common associations between ions and mitochondrial and fatty acid metabolism. a, b** Metabolites and proteins were sorted on their correlation with individual metals in the liver, then metabolite or gene -set enrichment analyses (MSEA, (**a**), GSEA, (**b**)) were performed. For each metal, the top three gene or metabolite sets are pictured. FDR-corrected $p$-values.**Adj.$P < 0.01$, *Adj.$P < 0.05$, 'Adj.$P < 0.1$ (**c**) Network representation, combining the most significant correlations between metals, between metals and phenotypes (Spearman Rho and limma test, both), as well as the metabolite and gene set enrichment analyses. All tests were two-sided when applicable and $p$-values are FDR corrected. To limit the number of nodes, only nodes with very significant associations are pictured (Adj.$P < 0.01$), but all edges with Adj.$P < 0.05$ are drawn. Source data are provided as a Source Data file.

either analyzed in positive (acidic solvent) or negative (basic solvent) ionization mode. GC-MS was performed on bistrimethyl-silyl-triflouroacetamide derivatized samples in a 5% phenyl GC column.

**Proteomics**. Proteomics was performed as described in Yu et al.[48]. Briefly, total proteome was isolated using a RIPA-M buffer (pH 7.5; nonidet NP-40, 0.1% sodium doexycholate, 150 mM NaCl, 1 mM EDTA, 50 mM Tris, protease inhibitor cocktail, 10 mM NaF, 5 mM 2-glycerophosphate) and lysed with a 8 M urea buffer (pH 8.1, 75 mM NaCl, 10 mM NaF, 5 mM 2-glycerophosphate; 8 M urea; protease inhibitor cocktail). Tissues were ground using a pestle grinder in the Ripa-M buffer before the samples were lysed in the urea buffer. After proteins were isolated, samples were trypsin digested overnight in the dark (~22 h @ 37 °C in a mild shaker) in a urea buffer with dithioethreitol and indole-3-acetic acid. After trypsinization, samples were passed through a C18 column and eluted in a 2% ACN solution with 0.1% FA. Samples were then run on a TripleTOF 5600 in SWATH mode. The search library used was generated from that of an earlier study[49]. For the data processing, we removed decoy peptides and peptides mapping only to reverse proteins, then summed raw intensities over different charge states and chemical modifications. The resulting peptide-level intensities were log$_2$ transformed and normalized by robust linear regression normalization, separately for each organ. We then removed peptides that are identified in only one sample, as well as proteins identified by a single unique peptide sequence. Finally, peptide-level intensities were summarized to the protein level with the MsqRobSum R package[50], (proteins were grouped by gene annotations).

**Data analysis**. Outlier measurements were removed using the 'normal' method (differences between each value and the mean divided by standard deviation) through the R package 'outliers'. Less than 5% of measurements were removed this way. All correlation analyses were performed by parallel linear regression using the 'limma' R package using age as a covariate[51]. $P$-values were corrected for multiple testing through the Benjamini–Hochberg false discovery rate method. Pearson correlation coefficients were computed separately through the 'cor.test' function. Age-corrected values were computed by estimating the effect of age through a linear model value ~age and extracting the intercept + residuals of the model. Those values were used for visualization only, but statistical analyses were performed using age as a covariate instead.

**MSEA**. In order to use the most complete and up-to-date repository of metabolite information, we based our analysis on the Human Metabolome Database (HMDB). We downloaded the database and parsed information about biological processes and cellular components from the relevant entries in the database, then generated a search file for the ClusterProfiler R package[52]. For MSEA, metabolites were sorted based on their degree of correlation with each metal (calculated with limma package, see 'analysis' section) using the following formula: $-\log_{10}(P\text{-value})*$ Pearson correlation coefficient. Metabolite Set enrichment analysis (MSEA) was performed using the ClusterProfiler package in R software.

**GSEA**. Gene set databases were downloaded from the Molecular Signatures Database (MSigDB) v7.2 through the MSigDRr R package. Similar to GSEA, all

proteins were sorted by $-\log_{10}(P\text{-value})*$ Pearson correlation coefficient and GSEA was performed with the ClusterProfiler R package.

**Networks**. All network representations show the correlation analyzes and MSEA/GSEA results outlined above. Only correlations with Adj. $P$-values < 0.005 were drawn. Network visualizations were generated using the igraph and ggraph R packages.

**Reporting summary**. Further information on research design is available in the Nature Research Reporting Summary linked to this article.

## Data availability

Raw metallomic data are included in this study as supplementary data 1. Proteomics has been submitted to the PRIDE repository under the identifier PXD011142[53]. Metabolomics data have been previously published and are available in Houtkooper et al.[39]. In addition, source data files are provided for every figure panel.

## Code availability

Data analysis was performed in R software version 3.5.2. The following open-source packages were used in analysis and figure generation: MsqRobSum: proteomics; limma: differential analyzes and parallel linear regressions; ClusterProfiler: Gene and Metabolite set enrichment analyzes; ggplot2: figure generation; igraph, ggraph: network visualizations. All code used in the paper is available on request.

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

## Acknowledgements

We thank all members of J. Auwerx's laboratory for helpful discussions. This project has received financial support from the CNRS through the MITI interdisciplinary programs, the Bullukian Foundation, the École Polytechnique Fédérale de Lausanne (EPFL), European Research Council (ERC-AdG-787702) and Swiss National Science Foundation (31003A_179435) and a Global Research Laboratory grant from the National Research Foundation of Korea (2017K1A1A2013124). L.S. was supported by EPFL and ENS de Lyon convention thanks to P. Gillet. L.J.E.G. was supported by the European Union Horizon 2020 Marie Skłodowska-Curie Individual Fellowship "AmyloAge" (896042).

## Author contributions

V.B. and J.A. conceived the project. R.H performed mouse clinical phenotyping and tissue collection. L.S. performed the metallomic measurements. R.A. and E.W. generated the proteomics dataset. J.D.M., V.B., and L.J.E.G analyzed the data with the help of L.S. and P.J.; V.B and J.D.M. wrote the manuscript with comments from all authors.

## Competing interests

The authors declare no competing interests.
