## [Peer Review File · Nature Communications]

REVIEWER COMMENTS

Reviewer #1 (Remarks to the Author):

The manuscript from Morel J.D. and coauthors on “The mouse metallomic landscape of aging and metabolism” represents an interesting contribution on the role of metal analysis and isotope ratio measurements in the metabolic changes associated to age. The measurement of total elemental concentrations in specific tissues using ICP-MS is relatively well established in the analytical field. But the correlation of such measurements with the isotopic signatures of two key-metabolic elements such as Cu and Zn, and, furthermore, the incorporation of the results of a set of metabolites and (few) genes relevant for specific protein biosynthesis, increases the quality and the broad interest of the work. Thus, I consider that the topic possesses enough novelty and the manuscript sufficient scientific quality to be considered as a contribution in Nature Communications. However, there are some specific aspects, particularly regarding to some elucubrations with respect to the relevance of the obtained data on certain metabolic pathways that the authors should sort out. Therefore, the paper could be considered after some major revisions focused on the points given below:

1. Abstract, last line. “modulating processes using metallome manipulation” I think this comment should be carefully put into perspective or eliminated. The incorporation pathway through diet of some of the elements analysed here (e.g. Fe) is tightly regulated by the intestinal absorption since not active metabolism or excretion exists. Thus, I think that metallome manipulation (if feasible!) could provide a lot of undesired consequences into the organisms. The abstract should make the manuscript attractive but realistic.
2. Line 47: the metallome is not just the “set of inorganic elements in an organism”. Please refer to the IUPAC definition of metallomics as the discipline and the metallome as the subject of the discipline (Pure App. Chem., 82, 2, 493-504, 2010).
3. Line 70: isotopic signature on Zn “distinct in kidney where glomerular filtration drives isotopic fractionation” this needs to be supported by experimental data or adequate references.
4. Line 72: “Cu has a unique concentration – isotopic composition signature for each organ” which is very different in the case of Zn. Is this driven by the route of entrance into the organism? Or the elimination pathway? The authors should find the justification for this (interesting) finding merely than the pure description. This is occurring all along the manuscript and the work could profit of a deeper analysis from a specialist in physiology.
5. Line 78: according to the results of Fig. S1C, Se shows a very similar concentration in the analyzed organs (brain, kidney and liver) so it is not heterogeneously distributed within the organs. In addition, this element is part of many important proteins in the organism thus I would not agree on its claimed restricted biological role. It cannot be compared with the role of Mo or Co.

6. Line 90: too speculative to assume that the obtained results of element accumulation in the brain precede the onset of neurodegenerative diseases. Have they conducted any additional test on the animal to prove such conclusion?

7. Line 91: why has been Rb selected for this study? Just for analogy to K? But the author monitor K as well, so I do not see the added information of this element. In particular, when the authors claim (out of Rb data) a disrupted K metabolism in old mice. This whole correlation needs to be well justified also with the corresponding references.

8. Line 108: How can Ca be negatively correlated with Mg and P (commonly found in association to phosphates) in most organs? Is this correct?

9. Line 141: When muscle Fe is higher, lower ROS production is observed. The opposite should be expected and also correlated with age since Fe is a potent ROS inductor.

10. Line 177: "similar roles of K and Rb which strengthen Rb as a proxy for K metabolism" I consider that the evidences found in the manuscript are not sufficient for proving this assumption.

Reviewer #2 (Remarks to the Author):

This paper is focused on the novel and exciting area of the mouse metallome (inorganic elements) and aging. The novelty is high because the metallome is not a mainstream platform like the metabolome, transcriptome, and proteome. The authors provide a comprehensive analysis of the metallome and its interaction with other omics in organs of aging mice. The main findings are:

1. The metallome fingerprints are organ-specific and age-dependent. This finding is interesting, though somewhat incremental/confirmatory as others have previously shown this to be true (PMID 32323920).
2. The metallome (kidney and liver) is associated with several cardiometabolic phenotypic traits.
3. Within liver, there are some interesting common associations between liver metal ions and chemical classes.

On page 4, the authors state that gastrocnemius muscle and heart muscle are devoid of ultra-trace elements, likely due to their limited involvement in biochemical functions. The authors should clarify what they mean by this sentence. If they mean to say that these tissues are not biochemically active, then that could be misinterpreted.

An interesting aspect of this work is the correlation of the metallome and isotope composition with metabolic health and various cardiometabolic traits. Of all of the tissues investigated, significant correlations between metals and metabolic phenotypes were only evident in liver and kidney. It is puzzling, not that liver and kidney metals were correlated with phenotypes, but that muscle and heart were NOT. The fact that cardiac / skeletal muscle metals did not correlate with strong age-related phenotypes within the same tissues needs some explanation. For example, glucose disposal following IP injection is primarily ascribed to skeletal muscle glucose uptake. Similarly, the main contributors to whole-body VO₂ peak are cardiac and skeletal muscle, but analyses showed that only metals in kidney and liver (not muscle) associate with whole-body cardiorespiratory fitness. If the metallome is truly influencing these metabolic phenotypes, then it is hard to reconcile that associations are not apparent within the most relevant tissues.

Given the dominant focus on liver metallomics and integration with metabolome and proteome, it is a shortcoming that there were no measurements of mitochondrial metabolism / ROS production in this tissue. Why was this data provided for skeletal muscle but not other tissues such as liver?

The authors should discuss how the metallome of postmitotic tissues (e.g., brain and muscle) may differ from other tissues and relevance to aging.

Reviewer #3 (Remarks to the Author):

In this manuscript, Morel and colleagues investigated the impact of physiological aging on the levels of inorganic bioessential elements, which they term “metallome”, in mice. They measured the levels of 14 inorganic elements, as well as the stable isotope composition for copper and zinc in 5 organs (brain, heart, kidney, liver and muscle) across three age groups (6-, 16- and 24-months old). The authors first defined organ-specific signatures of the metallome and isotope compositions and then analysed age-related differences. Organ-specific metallome signatures of aging could be identified in each of the investigated organs, with a notable accumulation of metals in the brain especially Fe and Cu. This is in line with previous observations describing accumulation of such metals in neurodegenerative disorders. Next, the authors compared their analysis to a recently published study that also investigated changes in the metallome in aging mice (Zhang, Podolskiy, et al. Aging Cell 2020). The analysis showed high level of consistency between the studies in term of organ-specific signatures, age-effects and correlation between elements. In the final part of the manuscript, the authors integrated the metallome analysis with phenotypic analysis of mice (across all age groups), as well as proteomics and metabolomics measurements of the same organs. The latter were restricted to the two extreme age groups: 6- and 24-months old. By these analyses, the

authors identified correlations between metals levels / isotope proportions and phenotypic traits, or metabolomic and proteomic signatures.

The analytical methods applied, and the results obtained are robust, as also highlighted by the reproducibility across independent studies. The manuscript is well written, and the presentation of the data is clear. My major concerns are on the novelty of the findings, given that similar analysis were recently reported in Zhang, Podolskiy, et al. *Aging Cell* 2020, and that the some of the conclusions of the authors appears to be over-stated based on the presented data.

Major criticism:

- In the Perspective paragraph the authors state that: The correlations between the metallome and other omic layers inform about the role of metals throughout biological networks, and provide new insights about organellar function in aging. I found this an over-statement. Given the exclusive correlative nature of the analysis presented, it is impossible to discern the role of metals in these biological processes in absence of perturbation experiments. As a demonstration of this, the authors rely on previous observations in other systems (e.g., line 136: "...increased Cu concentration and isotope composition have been previously associated with metabolic activity and growth of tumors...") or purely speculative statements to interpret their data (e.g., line 141: "Kidney iron also correlated with respiratory capacity (VO₂ increase, Fig. 3C) reflecting the importance of iron signaling in kidney erythropoietin production, a critical regulator of hemoglobin and erythrocyte production.").

- A perturbation experiment involving element supplementation or restriction/chelation followed by phenotypic and omics characterisation would be required to validate (as a proof of concept) at least one of the predictions made by the authors.

Minor criticism:

- It is not clear whether all the omics data were obtained from the very same mice used for metallome analysis? For the phenome is clear based on analysis shown in Figure 3, however for the other omics layers is less obvious. For instance, why metabolomics and proteomics analyses were not performed on all the animals used for metallome analysis? Can the authors please clarify this point?

- The authors should provide all the correlation values for individual proteins/metabolites and metals levels as supplementary tables.

- The authors should mention that accumulation of iron and other metals in the brain is a well-known feature of brain aging and not only neurodegenerative conditions. See for example: <https://pubmed.ncbi.nlm.nih.gov/6667718/> and <https://pubmed.ncbi.nlm.nih.gov/19563877/>

- Also relevant is that modulating iron levels by genetically altering the expression of master regulators of iron delivery is sufficient to revert (some) aging phenotypes in the brain: <https://bmcbiol.biomedcentral.com/articles/10.1186/s12915-017-0354-x>

Reviewer #1 (Remarks to the Author):

The manuscript from Morel J.D. and coauthors on “The mouse metallomic landscape of aging and metabolism” represents an interesting contribution on the role of metal analysis and isotope ratio measurements in the metabolic changes associated to age. The measurement of total elemental concentrations in specific tissues using ICP-MS is relatively well established in the analytical field. But the correlation of such measurements with the isotopic signatures of two key-metabolic elements such as Cu and Zn, and, furthermore, the incorporation of the results of a set of metabolites and (few) genes relevant for specific protein biosynthesis, increases the quality and the broad interest of the work. Thus, I consider that the topic possesses enough novelty and the manuscript sufficient scientific quality to be considered as a contribution in Nature Communications. However, there are some specific aspects, particularly regarding to some elucubrations with respect to the relevance of the obtained data on certain metabolic pathways that the authors should sort out. Therefore, the paper could be considered after some major revisions focused on the points given below:

We thank the reviewer for his encouraging comments.

1. Abstract, last line. “modulating processes using metallome manipulation” I think this comment should be carefully put into perspective or eliminated. The incorporation pathway though diet of some of the elements analysed here (e.g. Fe) is tightly regulated by the intestinal absorption since not active metabolism of excretion exists. Thus, I think that metallome manipulation (if feasible!) could provide a lot of undesired consequences into the organisms. The abstract should make to manuscript attractive but realistic.

We agree with the reviewer. We toned down the sentence which now reads "... potentially modulating cellular processes using careful and selective metallome manipulation" (line 44).

2. Line 47: the metallome is not just the “set of inorganic elements in an organism”. Please refer to the IUPAC definition of metallomics as the discipline and the metallome as the subject of the discipline (Pure App. Chem., 82, 2, 493-504, 2010).

We think that the format of *Nat. Comm.* does not allow to extend the text for a full definition of the metallome and metallomics. We, however, add the reference for further readings on this topic. We hope that this will match what the reviewer expected from us.

3. Line 70: isotopic signature on Zn “distinct in kidney where glomerular filtration drives isotopic fractionation” this needs to be supported by experimental data or adequate references.

We agree and toned down the sentence by adding " probably drives isotopic fractionation through a distillation process " (line 74) and a relevant reference (Jaouen et al. Dynamic homeostasis modeling of Zn isotope ratios in the human body. *Metallomics* 10.1039/c8mt00286j)(lines 72-73).

4. Line 72: “Cu has a unique concentration – isotopic composition signature for each organ” which is very different in the case of Zn. Is this driven by the route of entrance into the organism? Or the elimination pathway? The authors should find the justification for this (interesting) finding merely than the pure description. This is occurring all along the manuscript and the work could profit of a deeper analysis from a specialist in physiology.

We reworked the sentence which now reads "Copper isotope fractionation is more intense than that of Zn. Cu exhibits a unique concentration vs isotopic composition signature for each organ, depending on the occurrence of its oxidation state, Cu(II) compounds being isotopically heavier than Cu(I) compounds" (lines 74-77) and added a reference to support the claim (Albarède et

al. *Medical Applications of Isotope Metallomics. Reviews in Mineralogy and Geochemistry* 10.2138/rmg.2017.82.20)(lines 73-76).

5. Line 78: according to the results of Fig. S1C, Se shows a very similar concentration in the analyzed organs (brain, kidney and liver) so it is not heterogeneously distributed within the organs. In addition, this element is part of many important proteins in the organism thus I would not agree on its claimed restricted biological role. It cannot be compared with the role of Mo or Co.

This sentence has been reworked and now reads "Se for selenoproteins, Co for vitamin B12, Mo for molybdoenzymes, and Cd with unknown biological role⁴, exhibit a more heterogeneous distribution in the body than major elements and vary with age" (lines 81-83).

6. Line 90: too speculative to assume that the obtained results of element accumulation in the brain precede the onset of neurodegenerative diseases. Have they conducted any additional test on the animal to prove such conclusion?

Reviewer 3 has also made a comment on this point, but he highlights that this is not a new finding. He gave four references on this topic that we have added in the new text, which now reads "We find an age-dependent accumulation of Fe and Cu in healthy brain (Fig. 1F), which complements previous observations^{21–24} in the mouse. As high levels of transition metals such as Fe, Cu and Zn are known to be associated with amyloid- β plaques and α -synuclein accumulation in neurodegenerative diseases^{25,26}, all these results may suggest that metal accumulation precedes the formation of the protein aggregates" (lines 92-96).

7. Line 91: why has been Rb selected for this study? Just for analogy to K? But the author monitor K as well, so I do not see the added information of this element. In particular, when the authors claim (out of Rb data) a disrupted K metabolism in old mice. This whole correlation needs to be well justified also with the corresponding references.

Counterintuitively, Rb is an element that is present in relatively high amount, far more than Sr for instance that is the Ca minor element analog. Minor elements are often more sensitive to variations than major elements and can be used to monitor their metabolism, which was supported by the reference Fieve, R. R. *et al.* Rubidium: biochemical, behavioral, and metabolic studies in humans. *Am. J. Psychiatry* 10.1176/ajp.130.1.55.

8. Line 108: How can Ca be negatively correlated with Mg and P (commonly found in association to phosphates) in most organs? Is this correct?

This is correct and indeed surprising, but this is a robust pattern that is also present in the study of Zhang *et al.* (see figure S4).

9. Line 141: When muscle Fe is higher, lower ROS production is observed. The opposite should be expected and also correlated with age since Fe is a potent ROS inductor. The reviewer is correct, ROS in muscle indeed increases with age in our data (see Houtkooper, et al. Sci Rep 2011; Figure S1I, which used the same animals, pasted below for ease of reading). but we would like to stress here that all the phenotypes were corrected for age effects in the manuscript. The correlation between muscle Fe and ROS was not statistically significant (see Figure S5).

10. Line 177: “similar roles of K and Rb which strengthen Rb as a proxy for K metabolism” I consider that the evidences found in the manuscript are not sufficient for proving this assumption.

We have toned down the sentence which now reads "similar roles of K and Rb which strengthen the postulated status of Rb as a proxy for K metabolism" with the reference: Fieve, R. R. *et al.* Rubidium: biochemical, behavioral, and metabolic studies in humans. *Am. J. Psychiatry* 10.1176/ajp.130.1.55 (lines 192-193)

Reviewer #2 (Remarks to the Author):

This paper is focused on the novel and exciting area of the mouse metallome (inorganic elements) and aging. The novelty is high because the metallome is not a mainstream platform like the metabolome, transcriptome, and proteome. The authors provide a comprehensive analysis of the metallome and its interaction with other omics in organs of aging mice. The main findings are:

1. The metallome fingerprints are organ-specific and age-dependent. This finding is interesting, though somewhat incremental/confirmatory as others have previously shown this to be true (PMID 32323920).
2. The metallome (kidney and liver) is associated with several cardiometabolic phenotypic traits.
3. Within liver, there are some interesting common associations between liver metal ions and chemical classes.

We thank the reviewer for these kind comments.

On page 4, the authors state that gastrocnemius muscle and heart muscle are devoid of ultra-trace elements, likely due to their limited involvement in biochemical functions. The authors should clarify what they mean by this sentence. If they mean to say that these tissues are not biochemically active, then that could be misinterpreted.

We have rewritten this sentence which now reads “The gastrocnemius striated muscle and the myocardial muscle are depleted in ultra-trace elements, probably reflecting their limited involvement in specific biochemical synthesis” (lines 84-86).

An interesting aspect of this work is the correlation of the metallome and isotope composition with metabolic health and various cardiometabolic traits. Of all of the tissues investigated, significant correlations between metals and metabolic phenotypes were only evident in liver and kidney. It is puzzling, not that liver and kidney metals were correlated with phenotypes, but that muscle and heart were NOT. The fact that cardiac / skeletal muscle metals did not correlate with strong age-related phenotypes within the same tissues needs some explanation. For example, glucose disposal following IP injection is primarily ascribed to skeletal muscle glucose uptake. Similarly, the main contributors to whole-body VO₂ peak are cardiac and skeletal muscle, but analyses showed that only metals in kidney and liver (not muscle) associate

with whole-body cardiorespiratory fitness. If the metallome is truly influencing these metabolic phenotypes, then it is hard to reconcile that associations are not apparent within the most relevant tissues.

The reviewer is right, this has puzzled us too. The point is that we found no association between metals in muscle and heart and phenotypes at all. We believe this to be mostly an issue of underpowered statistics, rather than an interpretable result. In general, a lack of statistically significant association cannot be taken as evidence of the lack of a biological association, which is why we did not comment on it. In this case, we think the combined variation of the phenotype and muscle metallome data was too high, and with the correction for multiple testing we find no significant associations at all. We would also like to stress that the phenotypic variations were corrected for the effect of age which overrides many associations.

Regarding the correlation we find between IPGTT AUC and liver metals, it is accompanied by alterations of body weight and fat mass, which suggest a general obesity/diabetic-like phenotype. It could be that this obesity phenotype drives alteration in the liver, as is the case in human patients with metabolic syndrome, who often develop NAFLD. These liver alterations would in turn change metal concentrations in the liver. In other words, the metal alterations would be a consequence (or biomarker) of the phenotype rather than its cause. To represent this shift in causality, we have added the following sentence in the paragraph on phenotypes: “Importantly, these liver metal concentrations may not be causal in metabolic fitness, but rather represent a biomarker of liver health in metabolically healthy animals, as opposed to mildly obese or diabetic animals which develop liver dysfunction.”

Given the dominant focus on liver metallomics and integration with metabolome and proteome, it is a shortcoming that there were no measurements of mitochondrial metabolism / ROS production in this tissue. Why was this data provided for skeletal muscle but not other tissues such as liver?

Liver ROS were already measured in the paper, and can be seen on figure S4. After the automatic removal of 2 strong outliers, liver ROS did not significantly associate with metals, which is why they are not pictured on Figure 3.

As suggested by the reviewer, we have now analyzed associations between the metallome and mitochondrial metabolism in the liver. We used data on mitochondrial parameters that were measured in the same animals and previously published in Houtkooper, R., Argmann, C., Houten, S. et al. The metabolic footprint of aging in mice. Sci Rep 1, 134 (2011). Figure 3 has now been updated with the new parameters (also pasted below for ease of reading).

These new results are now commented in the text with the following sentences:

- “Iron and $\delta^{65}\text{Cu}$ are further associated with an increase in the activity of mitochondrial complex I and a reduction of the mtDNA/nuclear DNA ratio in the liver, suggesting that liver mitochondrial activity is associated to these changes in metal levels.” (lines 142-145).
- “In addition, liver calcium was strongly negatively correlated with the activity of mitochondrial complex IV. Calcium has been shown to bind and inhibit Mitochondrial complex IV *in vitro*³⁹, but these results suggest that this interaction may also be relevant *in vivo*.” (lines 153-156).

The material and methods section “In vivo phenotyping” section has also been updated with the relevant methods

The authors should discuss how the metallome of postmitotic tissues (e.g., brain and muscle) may differ from other tissues and relevance to aging.

The reviewer is correct that brain and muscle contain many non-dividing, post-mitotic cells, although the whole brain and whole muscle both contain a large diversity of cells, some of which are dividing. One major feature of age-effects in such tissues is the accumulation of heterogeneous aggregates of protein, lipids and metals. This is something we touch upon when we mention amyloid aggregates in the brain. Apart from this, we found no striking difference in aged-dependent metal accumulation between muscles and brain on the one hand and liver and kidney on the other, so we are unsure on how to conclude on this point.

Reviewer #3 (Remarks to the Author):

In this manuscript, Morel and colleagues investigated the impact of physiological aging on the levels of inorganic bioessential elements, which they term “metallome”, in mice. They measured the levels of 14 inorganic elements, as well as the stable isotope composition for copper and zinc in 5 organs (brain, heart, kidney, liver and muscle) across three age groups (6-, 16- and 24-months old). The authors first defined organ-specific signatures of the metallome and isotope compositions and then analysed age-related differences. Organ-specific metallome signatures of aging could be identified in each of the investigated organs, with a notable accumulation of metals in the brain especially Fe and Cu. This is in line with previous observations describing accumulation of such metals in neurodegenerative disorders. Next, the

authors compared their analysis to a recently published study that also investigated changes in the metallome in aging mice (Zhang, Podolskiy, et al. Aging Cell 2020). The analysis showed high level of consistency between the studies in term of organ-specific signatures, age-effects and correlation between elements. In the final part of the manuscript, the authors integrated the metallome analysis with phenotypic analysis of mice (across all age groups), as well as proteomics and metabolomics measurements of the same organs. The latter were restricted to the two extreme age groups: 6- and 24-months old. By these analyses, the authors identified correlations between metals levels / isotope proportions and phenotypic traits, or metabolomic and proteomic signatures.

The analytical methods applied, and the results obtained are robust, as also highlighted by the reproducibility across independent studies. The manuscript is well written, and the presentation of the data is clear. My major concerns are on the novelty of the findings, given that similar analysis were recently reported in Zhang, Podolskiy, et al. Aging Cell 2020, and that the some of the conclusions of the authors appears to be over-stated based on the presented data.

We thank the reviewer for these encouraging comments and would like to point out that the study of Zhang *et al.* only measures a fraction of the metals measured here, and no stable isotope compositions. Nor does it contain any omics data. The present study is the first to report an association study of the metallomic with other omic layers in the mouse.

Major criticism:

- In the Perspective paragraph the authors state that: The correlations between the metallome and other omic layers inform about the role of metals throughout biological networks, and provide new insights about organellar function in aging. I found this an over-statement. Given the exclusive correlative nature of the analysis presented, it is impossible to discern the role of metals in these biological processes in absence of perturbation experiments. As a demonstration of this, the authors rely on previous observations in other systems (e.g., line 136:” ...increased Cu concentration and isotope composition have been previously associated with metabolic activity and growth of tumors...” or purely speculative statements to interpret their data (e.g., line 141: “Kidney iron also correlated with respiratory capacity (VO₂ increase, Fig. 3C) reflecting the importance of iron signaling in kidney erythropoietin production, a critical regulator of hemoglobin and erythrocyte production.”).

We agree that this was an overstatement, and we have toned down this sentence which now reads “The correlations between the metallome and other omic layers inform about the involvement of metals throughout biological networks” (lines 201-203). In accordance with the first reviewer impression, we also have toned down the manuscripts in many places and we hope that the present revised version will match what the reviewer expected from us.

- A perturbation experiment involving element supplementation or restriction/chelation followed by phenotypic and omics characterisation would be required to validate (as a proof of concept) at least one of the predictions made by the authors.

The metabolic phenotypes and omics measured here represent subtle alterations that occur over an animal’s lifespan and which may inform about the interplay between metals and metabolic pathways in normal conditions. Manipulating the metallome to influence those pathways is an extremely interesting goal. However, as the first reviewer rightly states in his comments “*The incorporation pathway though diet of some of the elements analysed here (e.g. Fe) is tightly regulated by the intestinal absorption since not active metabolism of excretion exists. Thus, I think that metallome manipulation (if feasible!) could provide a lot of undesired consequences into the organisms.*” Therefore, while experiments of chelation or dietary manipulation of metals are interesting, they would have to be very carefully optimized to avoid undesired

toxicity and remain within physiological limits. Ultimately, we believe such experiments to be outside the scope of the present study, as they would extend its duration by several years. Indeed, obtaining the approval of the animal experimentation committee in Switzerland takes a minimum of six months. In addition, the metabolic alterations described here were observed in mice over 6 months of age and up to 24 months, and many metabolic dysregulations such as mild obesity or insulin resistance would not be detected in young mice.

Minor criticism:

- It is not clear whether all the omics data were obtained from the very same mice used for metallome analysis? For the phenome is clear based on analysis shown in Figure 3, however for the other omics layers is less obvious. For instance, why metabolomics and proteomics analyses were not performed on all the animals used for metallome analysis? Can the authors please clarify this point?

We agree with the reviewer that this was not stated clearly enough. Indeed, all omics data used in this paper came from the same animals. We have added the following sentence to clarify this “To examine the interplay between metallomics and other omics layers, we used previously published metabolomics⁴¹ and new proteomics analyses performed on liver samples from the same animals”(lines 162-164).

The reason for the different number of mice across layers (pictured on Fig 1A) is that the phenome, metabolome and proteome were measured prior to the metallome. Therefore, some samples were exhausted when we measured the metallome, but we limited our analysis to matching samples.

- The authors should provide all the correlation values for individual proteins/metabolites and metals levels as supplementary tables.

These are now provided in table S3.

- The authors should mention that accumulation of iron and other metals in the brain is a well-known feature of brain aging and not only neurodegenerative conditions. See for example: <https://pubmed.ncbi.nlm.nih.gov/6667718/> and <https://pubmed.ncbi.nlm.nih.gov/19563877/> This has been done, see point 6 of reviewer 1.

- Also relevant is that modulating iron levels by genetically altering the expression of master regulators of iron delivery is sufficient to revert (some) aging phenotypes in the brain: <https://bmcbiol.biomedcentral.com/articles/10.1186/s12915-017-0354-x>

We thank the reviewer for this interesting reference that we have added in the text.

REVIEWERS' COMMENTS

Reviewer #1 (Remarks to the Author):

The revised version of the manuscript from Morel J.D. and coauthors on “The mouse metallomic landscape of aging and metabolism” represents an interesting contribution on the role of metal analysis and isotope ratio measurements in the metabolic changes associated to age.

In this latest version, the authors have clarified the doubts raised in the previous manuscript and soften some of the assumptions claimed in the first version of the work. I consider that the paper could be accepted in the current form.

Reviewer #2 (Remarks to the Author):

The authors have adequately addressed all of my earlier concerns. In several cases, my queries were addressed on the basis of inadequate sample size, which remains a limitation of the work. No further comments.

Reviewer #3 (Remarks to the Author):

The authors have revised their manuscript in a satisfactory way to avoid overstatements and incorporate reviewers' suggestions. Since follow up experiments have been considered out of scope for this manuscript, I have no further comments and recommend publication of the revised manuscript.